# Rethinking RL Evaluation: Can Benchmarks Truly Reveal Failures of RL Methods?

## Abstract

Current benchmarks are inadequate for evaluating progress in reinforcement learning (RL) for large language models (LLMs). Despite recent benchmark gains reported for RL, we find that training on these benchmarks' training sets achieves nearly the same performance as training directly on the test sets, suggesting that the benchmarks cannot reliably separate further progress. To study this phenomenon, we introduce a diagnostic suite and the Oracle Performance Gap (OPG) metric that quantifies the performance difference between training on the train split versus the test split of a benchmark. We further analyze this phenomenon with stress tests and find that, despite strong benchmark scores, existing RL methods struggle to generalize across distribution shifts, varying levels of difficulty, and counterfactual scenarios: shortcomings that current benchmarks fail to reveal. We conclude that current benchmarks are insufficient for evaluating generalization and propose three core principles for designing more faithful benchmarks: sufficient difficulty, balanced evaluation, and distributional robustness.

## 1 INTRODUCTION

Reinforcement Learning (RL) has emerged as a powerful paradigm for post-training Large Language Models (LLMs), significantly enhancing their capabilities on complex, multi-step reasoning tasks (Ouyang et al., 2022). Methods based on Reinforcement Learning from Human Feedback (RLHF) and Direct Preference Optimization (DPO) (Rafailov et al., 2023) have become standard practice for aligning LLMs. These paradigms are often powered by foundational algorithms like Proximal Policy Optimization (PPO) (Schulman et al., 2017), with state-of-the-art variants such as Group Relative Policy Optimization (GRPO) (Shao et al., 2024) pushing models to achieve remarkable performance on benchmarks like GSM8K (Cobbe et al., 2021) and MATH (Hendrycks et al., 2021). These successes, often marked by state-of-the-art results (Lewkowycz et al., 2022; Lightman et al., 2023), are widely interpreted as a significant leap forward, suggesting that RL-based alignment is a key pathway toward developing more general and robust machine reasoning systems.

Despite impressive reported gains, a key question is whether current benchmarks still meaningfully assess generalization. Our findings suggest that the traditional assumption underlying benchmark design, that a model's ability to perform well on unseen test examples is sufficient to measure generalization, no longer holds for RL. We find that RL-based reasoning models trained on the training split achieve nearly the same performance as those trained directly on the test split, indicating that "unseen-ness" alone is no longer the challenging or discriminative criterion. This calls for the rethinking of evaluation: rather than relying solely on disjoint train/test splits, future benchmarks must incorporate settings that remain sensitive to deeper forms of generalization and can reveal weaknesses that simple data separation fails to expose.

To systematically investigate this phenomenon, we introduce a multi-faceted empirical framework designed not merely to measure performance, but to deconstruct it. We first propose a novel metric, Oracle Performance Gap (OPG), to provide a quantitative measure of the vanishing generalization gap observed in RL agents. Building on this, we subject RL-tuned models to a suite of rigorous stress tests to probe the fragility of their learned skills: Complexity Test to assess performance across varying complexity; Distribution Test to measure brittleness against out-of-distribution data; and Counterfactual Robustness Test that creates a direct conflict between memorized patterns and deductive reasoning. Our findings remain consistent across multiple RL algorithms and model scales,

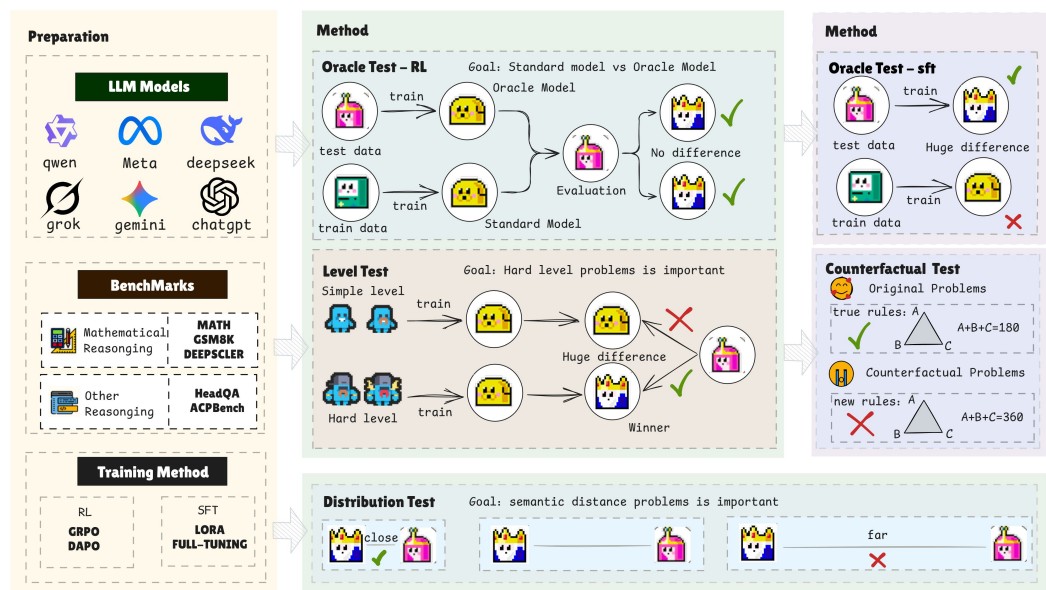

Figure 1: *Overview of our empirical framework.* The workflow begins by diagnosing benchmark flaws with novel metrics to uncover a core symptom: a vanishing generalization gap. It then proceeds through a suite of stress tests that reveal the brittle, shortcut-based nature of the learned skills, culminating in a new set of principles for more robust evaluation.

highlighting the pervasiveness of this issue. Overall, our contributions are:

❖ **Illusion of Capability.** We provide quantitative evidence that high scores on prevailing benchmarks may not be reliable indicators of true capability. We demonstrate that their structural limitations—evidenced by a vanishing generalization gap and failures in stress tests (e.g., OOD, counterfactual)—may reward brittle, non-generalizable behaviors, thus calling into question their reliability as measures of true reasoning.

❖ **Novel Diagnostic Framework.** We introduce a new diagnostic framework, including the OPG and various decay curves (difficulty, distributional, and counterfactual), to systematically probe and quantify the fragility of learned reasoning skills by RL.

❖ **Actionable Design Principles.** Based on our findings, we propose a set of actionable principles for designing next-generation benchmarks that can more robustly evaluate an agent's true, transferable reasoning abilities.

## 2 DIAGNOSING GENERALIZATGION BENCHMARKING WITH ORACLE PERFORMANCE GAP

The standard approach to evaluating LLM reasoning is to measure performance on a held-out test set, under the assumption that success on unseen data reflects generalization. To examine whether this assumption still holds for RL-based methods, we introduce a diagnostic framework that tests whether "unseen-ness", the common practice of relying on disjoint train/test splits, continues to provide a valid measure of generalization. Our framework compares RL models trained on training split with the Oracle model trained directly on the test split and finds that their performance is nearly identical, indicating that test-set "unseen-ness" alone has ceased to be a diagnostic signal of generalization.

### 2.1 ANALYSIS FRAMEWORK

#### 2.1.1 ORACLE PERFORMANCE GAP (OPG)

We introduce the Oracle Performance Gap (OPG) as a diagnostic metric to audit the validity of a benchmark. Unlike standard generalization gaps (which measure model overfitting), OPG measures

the *discriminative power* of the test set by comparing a standard model (trained on training data) against an "Oracle" model (fine-tuned explicitly on the test set).

Formally, for a given fine-tuning algorithm $\mathcal{A} \in \{SFT, RL\}$, let $P(M, \mathcal{D})$ denote the performance of a model $M$ on a dataset $\mathcal{D}$, measured by pass@1 accuracy. We define the OPG as the normalized performance difference:

$$\text{OPG}_{\mathcal{A}} \triangleq \frac{P(M_{\mathcal{A},\text{test}}, \mathcal{D}_{\text{test}}) - P(M_{\mathcal{A},\text{train}}, \mathcal{D}_{\text{test}})}{P(M_{\mathcal{A},\text{test}}, \mathcal{D}_{\text{test}})}. \tag{1}$$

With OPG, we establish an upper bound for performance via memorization and assess whether the benchmark effectively challenges algorithm $\mathcal{A}$. We distinguish two outcomes:

1. **Effective Generalization (OPG$_{\mathcal{A}} \gg 0$):** A significant gap confirms that the benchmark poses a valid challenge, as the test set contains specific patterns or difficulties that cannot be trivially inferred from the training set.
2. **Benchmark Failure (OPG$_{\mathcal{A}} \lesssim 0$):** A negligible gap indicates structural redundancy. It suggests the test set fails to differentiate between true generalization and simple pattern matching, as "seeing" the test data offers no performance advantage.

### 2.1.2 EXPERIMENT SETUP

Our analysis spans four benchmarks: MATH, GSM8K, HeadQA, and DeepScaler (Luo et al.). We use two base models from the Qwen family, `Qwen2.5-3B-Instruct` and `Qwen2.5-7B-Instruct`. To systematically isolate the effects of the fine-tuning paradigm and data distribution, we create and compare a suite of six model variants for each base model:

- **Baseline** ($M_{base}$)**:** The original instruction-tuned model without any additional fine-tuning.
- **Standard SFT** ($M_{SFT,train}$)**:** The base model fine-tuned on the official training set containing only standard question-answer pairs.
- **SFT with CoT** ($M_{SFT,formatted}$)**:** The base model fine-tuned on a formatted training set that includes detailed, teacher-generated chain-of-thought (CoT) reasoning steps.
- **RL on Train Set** ($M_{RL,train}$)**:** The base model fine-tuned on the official training set using GRPO, a state-of-the-art RL algorithm.
- **SFT Oracle** ($M_{SFT,test}$)**:** The base model fine-tuned directly on the *test set* using SFT. This serves as a practical upper bound for SFT performance on the test distribution.
- **RL Oracle** ($M_{RL,test}$)**:** The base model fine-tuned directly on the *test set* using the same GRPO setup. This provides an upper bound for the RL agent's ability to exploit the test set.

All models are evaluated on the official test sets using pass@1 accuracy. Full implementation details, hyperparameters, and evaluation protocols are provided in Appendix A.

Table 1: *Benchmark Limitation Illustrated by Qwen2.5 Model Performance.* The table is reorganized by benchmark, comparing performance across 3B and 7B model scales.

| Benchmark | Model Size | RL on Train Set Subsets (%) | | | | RL Oracle ($M_{RL,test}$) | Baseline ($M_{base}$) | OPG (%) |
|---|---|---|---|---|---|---|---|---|
| | | 10% | 20% | 50% | 100% | | | |
| MATH | 3B | $63.88_{\pm1.04}$ | $65.18_{\pm0.84}$ | $64.84_{\pm1.25}$ | $64.62_{\pm0.98}$ | $64.62_{\pm1.11}$ | 62.20 | 0.00 |
| | 7B | $73.64_{\pm0.48}$ | $73.28_{\pm0.68}$ | $73.04_{\pm0.91}$ | $74.04_{\pm0.39}$ | $74.00_{\pm0.68}$ | 68.80 | -0.05 |
| GSM8K | 3B | $82.95_{\pm0.38}$ | $83.93_{\pm0.47}$ | $86.93_{\pm0.34}$ | $87.04_{\pm0.49}$ | $87.98_{\pm0.35}$ | 83.02 | 1.07 |
| | 7B | $88.76_{\pm0.47}$ | $89.58_{\pm0.44}$ | $91.14_{\pm0.30}$ | $91.72_{\pm0.31}$ | $91.87_{\pm0.31}$ | 88.40 | 0.16 |
| DeepScaler | 3B | $34.12_{\pm0.94}$ | $32.68_{\pm1.02}$ | $35.75_{\pm0.96}$ | $35.22_{\pm0.89}$ | $34.95_{\pm0.91}$ | 33.38 | -0.77 |
| | 7B | $42.05_{\pm0.84}$ | $42.09_{\pm0.57}$ | $42.84_{\pm0.73}$ | $42.36_{\pm0.77}$ | $42.64_{\pm0.75}$ | 35.70 | 0.66 |
| HeadQA | 3B | $62.98_{\pm0.77}$ | $65.90_{\pm1.19}$ | $67.16_{\pm0.79}$ | $67.24_{\pm0.59}$ | $67.57_{\pm0.70}$ | 54.96 | 0.49 |
| | 7B | $72.94_{\pm0.90}$ | $72.90_{\pm0.79}$ | $74.39_{\pm0.60}$ | $75.20_{\pm0.66}$ | $75.60_{\pm0.50}$ | 52.24 | 0.53 |

Table 2: *SFT Performance Reveals the Expected Generalization Gap.* This table presents the SFT results organized by benchmark, with a direct comparison between the 3B and 7B model scales.

| Benchmark | Model Size | SFT Performance Metrics | | | OPG (%) |
|---|---|---|---|---|---|
| | | $M_{SFT,train}$ | $M_{SFT,test}$ | $M_{SFT,formatted}$ | |
| MATH | 3B | 17.20% | 40.00% | 31.02% | 22.45 |
| | 7B | 23.60% | 64.20% | 42.00% | 34.58 |
| GSM8K | 3B | 16.83% | 68.05% | 64.82% | 4.75 |
| | 7B | 19.71% | 79.04% | 75.36% | 4.66 |
| DeepScaler | 3B | 8.51% | 27.03% | 22.57% | 16.50 |
| | 7B | 12.57% | 36.76% | 23.51% | 36.04 |

## 2.2 RESULT

**Finding 1: The Vanishing Generalization Gap Shows Unseen-ness is an Insufficient Criterion.**
The OPG analysis reveals a stark contrast between SFT and RL paradigms, as detailed in Tables **??** and 2. While SFT models exhibit a large and expected OPG in a challenging generalization setting, this gap collapses to near-zero for RL-trained models. To rule out the concern that this is caused by data leakage in the base model, we verified that both our fine-tuned models significantly outperform the untrained baseline, confirming that this result reflects genuine training behavior.

Furthermore, to address the potential effect of the test set being much smaller than the training set, we trained RL models on various subsets of the training data. The OPG remained consistently low across all sizes, suggesting our conclusion is robust to the effects of data quantity. Together, these results provide strong evidence that the classical assumption—that performance on "unseen" test data is a sufficient measure of generalization—no longer holds for RL. This indicates that current benchmarks are unlikely to meaningfully assess future progress in RL generalization.

> **Takeaway ❶.** Our OPG analysis shows that RL models trained on the training split perform nearly identically to those trained directly on the test split, indicating that test-set "unseen-ness" is no longer a valid measure of generalization for RL.

## 3 DECONSTRUCTING GENERALIZATION AND DERIVING BENCHMARK PRINCIPLES

Building on our analysis in Section 2, which showed that simple unseen-ness is insufficient to measure generalization for RL, we next explore directions for designing benchmarks that go beyond this paradigm. We identify three key principles for effective evaluation of RL generalization: testing cross-difficulty generalization, assessing robustness to distributional shifts, and probing counterfactual reasoning.

### 3.1 THE DIFFICULTY TEST: GENERALIZATION ACROSS COMPLEXITY

By stratifying the benchmark by difficulty, we break apart the standard average aggregation an artifact of relying solely on "unseen-ness" and reveal how it conceals critical failures, pointing toward key principles for designing more informative generalization evaluations.

### 3.1.1 THE PARADOX OF AVERAGE SCORES

**Setup.** We conduct a cross-difficulty analysis by training five specialist models ($M_{L_i}$), each fine-tuned on a single difficulty partition of the MATH dataset ($\mathcal{D}_{\text{train}}^{L_i}$, see Appendix B.1 for the partition protocol). Each specialist is then evaluated on all five partitions, where a model's performance on its own training data serves as an oracle's performance benchmark against which its true generalization to unseen partitions is measured.

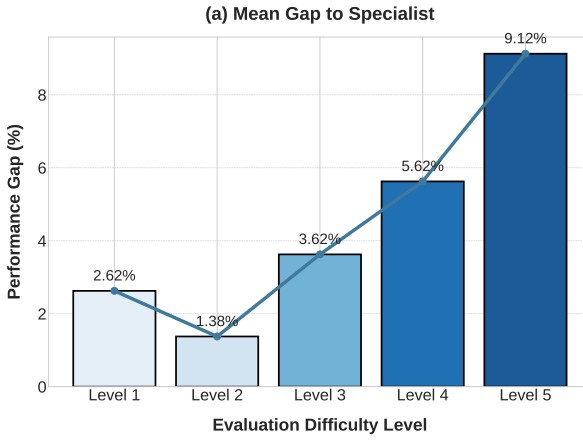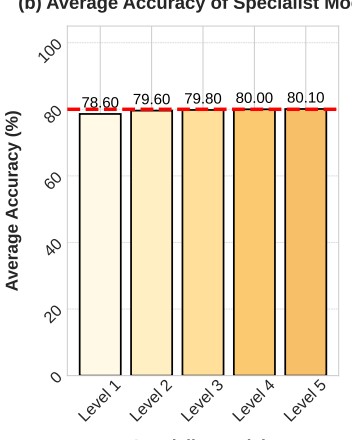

Figure 2: *The Illusion of Average Performance.* **(a)** The mean performance gap between the best (specialist) model and the average of all other models widens dramatically as task difficulty increases. **(b)** Surprisingly, the average scores of these specialists **(calculated across all five difficulty partitions)** are nearly identical. This contrast illustrates how a difficulty-agnostic evaluation can mask substantial differences in generalization capability. Full performance data is provided in Appendix B.2.

We observe that RL models trained on different difficulty levels exhibit markedly different degrees of generalization: models trained on harder levels transfer well to easier ones, while those trained on easier levels struggle to generalize to harder tasks, as shown in Figure 3. Figure 2(a) further demonstrates this asymmetry, where specialists trained on easier levels perform poorly on more difficult data. However, when evaluation follows the original benchmark protocol—computing a single average score over the entire difficulty-mixed test set (the standard "unseen-ness" setting)—these models achieve nearly identical aggregate results, as shown in Figure 2(b). This illustrates that difficulty-agnostic averaging masks meaningful differences in generalization and creates the misleading impression that these models are equally capable. In contrast, difficulty-aware evaluation, as in Figure 2(a), clearly separates models by their true generalization ability.

**Finding 2: A difficulty-aware train–test split provides an effective setting for evaluating generalization.** Our cross-difficulty evaluation empirically confirms the failure mode concealed by average score and reveals that difficulty-aware partition could be a more effective paradigm for generalization evaluation. This is evidenced by two key findings:

- **The Masking Effect is Confirmed:** In stark contrast, Figure 2(b) shows that these profound, growing differences in capability are completely masked by the final average scores. Paradoxically, the average scores for all specialist models remain nearly identical.
- **The Oracle Gap Re-emerges and Widens at the Micro-Level:** Contradicting the near-zero OPG found earlier, our micro-level analysis reveals a different reality. As shown in Figure 2(a), the mean performance gap to the specialist model ($M_{L_j}$)—which serves as the oracle for its specific difficulty level—is not zero. Instead, this difficulty-stratified oracle gap reappears and progressively widens as tasks become more complex. This demonstrates that the oracle gap is not truly gone, but merely hidden by aggregation.

> **Principle ❶.** Our findings suggest that a benchmark's evaluation protocol should be balanced and stratified by difficulty to prevent strong performance on easy problems from masking failures on complex tasks. Instead of relying on a single aggregate score, benchmarks should report performance across different difficulty levels separately. This provides a more faithful and granular assessment of a model's true reasoning capabilities and exposes weaknesses in generalization.

### 3.1.2 The Impact of Training Difficulty

**Setup and Phenomenon.** To isolate the impact of training data difficulty on generalization, we conduct a final analysis on the five "specialist" models. Our goal is to observe a specific phenomenon: whether a model's ability to generalize to unseen difficulties is a direct function of its training complexity. To measure this precisely, we introduce a new metric, *Average Cross-Difficulty Generalization* ($\bar{P}_{cross}$), defined as a model's average pass@1 accuracy on all difficulty levels *other* than its own ($j \neq i$). This metric is designed to purely capture transferable skill by excluding any in-distribution (data leakage) effects. The phenomenon we test for is whether this generalization score is a monotonically non-decreasing function of the training level's difficulty, formalized as:

$$\forall i, k \in \{1, \ldots, 5\} \text{ where } i > k, \quad \bar{P}_{cross}(M_{L_i}) \geq \bar{P}_{cross}(M_{L_k}) \tag{2}$$

**Finding 3: Training on Difficult Problems Boosts Transferable Generalization.** Our complexity test reveals a stark and telling pattern of asymmetric generalization, as illustrated in Figure 3. The analysis shows a clear, monotonic trend: the more difficult the training data, the better a model's Average Cross-Difficulty Generalization score becomes—a metric designed to purely measure transferable skill. Specifically, models trained on the highest difficulty levels (L4 and L5) consistently achieve the best average performance on unseen difficulty levels, exhibiting a uniformly superior performance profile compared to their counterparts trained on easier data (L1-L3), whose generalization capability remains notably limited. This demonstrates that mastering complexity is the key to instilling robust, transferable reasoning principles that apply broadly across different contexts. In contrast, an overabundance of easy training data appears to only teach narrow,

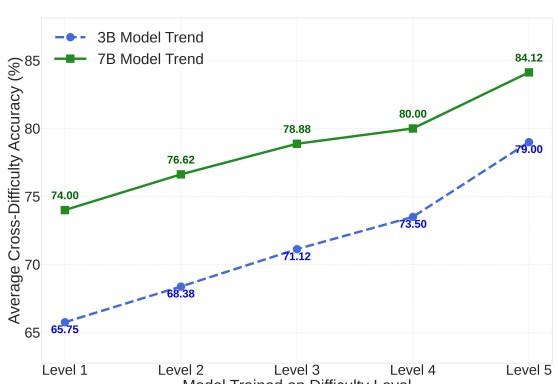

Figure 3: *The Average Cross-Difficulty Generalization score for 3B and 7B models.* The y-axis represents the average accuracy of a specialist model (trained on level $L_i$) on all other, unseen difficulty levels. Both models show a clear trend: as the training data complexity increases, the model's ability to generalize to other difficulties improves, with the Level 5-trained model being the strongest generalist.

context-specific solutions that fail to generalize. This finding therefore leads to a direct and actionable principle for building more capable agents: to foster the development of truly generalizable skills, the training curricula provided by benchmark suites must contain a significant proportion of high-complexity problems.

## 3.2 Preliminary evidence of potential benchmarking improvement

The previous section demonstrated that training on complex problems is essential for building robust models. However, difficulty is only one dimension of generalization. To further probe the limitations of current benchmarks and motivate our design principles, we now introduce two powerful stress tests designed to assess deeper forms of generalization. The Distribution Test quantifies a model's brittleness to semantic shifts in the data distribution (Section 3.2.1). The Counterfactual Test directly measures a model's ability to reason from novel premises versus merely reciting memorized knowledge (Section 3.2.2).

### 3.2.1 The Distribution Test: A Proposed Method for Quantifying Brittleness

**Setup.** To quantify generalization as a function of semantic distance, we mapped a corpus of 44,785 mathematics problems into a vector space using the `all-mpnet-base-v2` sentence encoder and partitioned the embeddings into three semantic clusters via K-Means ($k = 3$). From this clustered space, we constructed our core training and test sets, a process visualized in Appendix D.

- **Core Training Set ($\mathcal{D}_{\mathbf{core}}$):** We formed a highly concentrated and semantically narrow training set by selecting the 2,000 problems closest to a chosen cluster's centroid. This set is designed to represent a very narrow data distribution.

- **Core-Trained Model ($M_{\mathbf{core}}$):** A specialist model was then created by fine-tuning the base model exclusively on the $\mathcal{D}_{\mathrm{core}}$ dataset. This process, using our fine-tuning function $T$, is formally represented as $M_{\mathrm{core}} \triangleq T(M_{base}, \mathrm{RL}, \mathcal{D}_{\mathrm{core}})$. This model is intended to be an expert only on this narrow distribution.

Finally, five test sets, $\{\mathcal{D}_{\mathrm{test}}^{d_k}\}_{k=1}^5$, were constructed by sampling 80 problems each from the remaining data, which were binned according to increasing semantic distance $d_k$ from the $\mathcal{D}_{\mathrm{core}}$ centroid.

---

**Hypothesis 1: Localized Training Leads to Brittle Generalization and Performance Inversion**

We hypothesize that skills learned on a narrow distribution are brittle and fail to generalize to out-of-distribution (OOD) data. We measure this by analyzing the performance gain of the specialist model over the baseline, $\mathrm{Gain}(k) \triangleq P(M_{\mathrm{core}}, \mathcal{D}_{\mathrm{test}}^{d_k}) - P(M_{base}, \mathcal{D}_{\mathrm{test}}^{d_k})$, as a function of semantic distance $d_k$. This experiment tests for the existence of a performance inversion, a critical failure mode where the gain becomes negative for sufficiently distant data:

$$\exists k \in \{1, \ldots, 5\} \quad \text{s.t.} \quad \mathrm{Gain}(k) < 0 \tag{3}$$

Confirming this would demonstrate that fine-tuning can instill harmful, non-generalizable biases.

---

Table 3: *Performance on the Distribution Test.* With test sets denoted by $\mathcal{D}_{\mathrm{test}}^{d_k}$ where $d_k$ is the increasing semantic distance from the core training set. The Performance Gain row shows the fine-tuned model's advantage inverting into a penalty.

| Model / Metric | $\mathcal{D}_{\mathrm{test}}^{d_1}$ | $\mathcal{D}_{\mathrm{test}}^{d_2}$ | $\mathcal{D}_{\mathrm{test}}^{d_3}$ | $\mathcal{D}_{\mathrm{test}}^{d_4}$ | $\mathcal{D}_{\mathrm{test}}^{d_5}$ |
|---|---|---|---|---|---|
| $M_{core}$ (%) | 46.25 | 43.25 | 43.75 | 45.00 | 45.00 |
| $M_{base}$ (%) | 28.75 | 42.50 | 43.75 | 46.25 | 47.50 |
| **Performance Gain** (%) | **+17.50** | **+0.75** | **0.00** | **-1.25** | **-2.50** |

**Finding 4: Performance Inversion Motivates Distributional Robustness.** Our distribution test (Table 3) reveals that fine-tuning on a narrow data distribution can be actively harmful. While a specialized model ($M_{\mathrm{core}}$) excels on in-distribution data, its advantage vanishes with semantic distance, culminating in a performance inversion on the farthest OOD set. Here, the specialist's accuracy collapses below that of the un-tuned baseline, demonstrating that over-specialization can instill harmful biases that actively interfere with a model's general capabilities.

---

**Principle ❷.** Incorporating Distributional Robustness. Our findings suggest that a faithful benchmark should go beyond in-distribution evaluation to actively probe for robustness against distributional shifts. It should include a spectrum of out-of-distribution (OOD) challenges to penalize brittle, over-specialized models—particularly those prone to "performance inversion" where fine-tuning becomes actively harmful—and reward models with true, generalizable skills

---

### 3.2.2 THE COUNTERFACTUAL ROBUSTNESS TEST: REASONING VS. RECITATION

**Setup.** To rigorously test whether our models perform genuine deductive reasoning or merely recite pre-trained knowledge, we designed a counterfactual robustness test. The experiment is constructed around the following key components:

- **Test Sets ($\mathcal{D}_{bal}$ and $\mathcal{D}_{cf}$):** Our experiment uses two test sets: a standard balanced set, $\mathcal{D}_{bal}$, from the MATH benchmark, and our primary evaluation set, $\mathcal{D}_{cf}$, which is created by transforming a subset of problems from $\mathcal{D}_{bal}$ (see Appendix E.1 for details).

- **Counterfactual Transformation ($c_{\text{real}} \to c_{\text{fake}}$):** The transformation process involves identifying a problem's core, real-world mathematical rule, $c_{\text{real}}$, and explicitly replacing it with a novel, contrary-to-fact premise, $c_{\text{fake}}$.

- **Evaluation Criterion:** A model's response is marked as correct only if it correctly and exclusively applies the explicitly stated counterfactual premise, $c_{\text{fake}}$. This strict criterion ensures we are measuring on-the-fly reasoning rather than answer correctness based on memorized knowledge.

We then evaluated our main RL-tuned models, `Qwen2.5-3B-MATH` and `Qwen2.5-7B-MATH`, on the new counterfactual set $\mathcal{D}_{cf}$.

---

**Hypothesis 2: Models Prioritize Recitation Over Reasoning**

We test the hypothesis that models default to reciting memorized knowledge ($c_{\text{real}}$) instead of reasoning from a novel premise ($c_{\text{fake}}$). A confirmation is indicated by a significant performance collapse on the counterfactual set, formalized as:

$$P(M_{\text{RL},train}, \mathcal{D}_{cf}) \ll P(M_{\text{RL},train}, \mathcal{D}_{bal}) \tag{4}$$

---

**Finding 5: Counterfactual Failures Reveal Recitation Over Reasoning.** Our counterfactual robustness test reveals a critical failure in models' ability to reason from novel premises. This is quantitatively evident in the severe performance degradation on the counterfactual test, where accuracies for our 7B and 3B models drop from 74.8% and 64.2% to 41.2% and 36.0%, respectively (Table 4). A qualitative analysis of the model's chain-of-thought process confirms the cause of this failure (see Appendix E.2 for a detailed example). When presented with a problem that redefines the order of operations to PESAMD, the model completely disregards the new rule and defaults to the standard PEMDAS operations it has memorized. This provides definitive evidence that it operates as a pattern-matching engine that recites knowledge, rather than as a flexible, deductive reasoner.

Table 4: Performance collapse.

| Model | $\mathcal{D}_{bal}$ (%) | $\mathcal{D}_{cf}$ (%) |
|---|---|---|
| 3B-MATH | 64.2 | 36.0 |
| 7B-MATH | 74.8 | 41.2 |

---

**Principle ❸:** Assessing Counterfactual Reasoning. A faithful benchmark requires distinguishing true deduction from mere recitation. Our counterfactual test highlights a critical failure mode: when faced with novel, contrary-to-fact rules, models consistently default to reciting memorized knowledge rather than applying the new premise. Consequently, to penalize this brittle behavior and reward flexible reasoning, effective evaluation entails including problems that create a direct conflict between memorized priors and on-the-fly deduction.

---

## 4 RELATED WORK

### 4.1 REASONING IN LARGE LANGUAGE MODELS

The advent of Chain-of-Thought (CoT) prompting has become a cornerstone for eliciting complex reasoning in Large Language Models (LLMs) (Wei et al., 2022). This approach, along with advanced strategies like Tree of Thoughts (Wang et al., 2022; Yao et al., 2023; Yu et al., 2025), improves reasoning by guiding models to generate step-by-step rationales. Alongside prompting, fine-tuning on high-quality reasoning datasets remains a critical method for instilling these skills directly into model parameters (Lewkowycz et al., 2022). These parallel research thrusts have driven remarkable performance improvements on popular reasoning benchmarks. However, our work diverges from this trend of score optimization. Instead, we critically interrogate the reliability of the benchmarks themselves and argue that the resulting performance gains are often an illusion created by benchmark flaws, rather than a sign of true reasoning acquisition.

### 4.2 REINFORCEMENT LEARNING FOR LLM ALIGNMENT

To overcome the limitations of passive Supervised Fine-Tuning (SFT), which is confined to imitating a static dataset, Reinforcement Learning (RL) is increasingly used to actively explore and

optimize LLMs for reasoning tasks by directly rewarding correct outcomes (Ouyang et al., 2022). This approach, often implemented with foundational algorithms like PPO (Schulman et al., 2017), is highly effective at increasing benchmark scores. Building on this, state-of-the-art algorithms such as Group Relative Policy Optimization (GRPO) have emerged to further enhance training stability by comparing a chosen response against a cohort of rejected ones, providing a more robust learning signal (Shao et al., 2024). However, the reliance on outcome-based rewards has sparked a key debate, with researchers advocating for more robust process-based reward modeling, which evaluates the reasoning steps themselves to prevent models from "lucking into" correct answers (Li et al., 2025). Our work adds a critical dimension to this discussion by demonstrating that even with advanced outcome-based RL methods, the structural flaws of the underlying benchmark can lead to the reinforcement of brittle, non-generalizable behaviors.

## 4.3 ANALYSIS AND CRITIQUE OF BENCHMARKS

While benchmarks like GSM8K (Cobbe et al., 2021) and MATH (Hendrycks et al., 2021) are vital for driving progress, a growing body of work shows that models often exploit dataset artifacts and "shortcuts" rather than learning the intended skill (Geirhos et al., 2020). This has led to more rigorous evaluation methods designed to probe for true generalization, such as testing on out-of-distribution (OOD) or adversarially perturbed examples (Jia & Liang, 2017). However, these methods primarily stress the model's capabilities, often without diagnosing the underlying benchmark properties that permit brittle learning. Our research contributes to this line of critical analysis by introducing a novel set of diagnostic tools that analyze the structural properties of benchmarks themselves. Specifically, our Oracle Performance Gap (OPG) and difficulty-stratified evaluations provide quantitative evidence that high scores can be an illusion of capability, motivating the principles we propose for designing more reliable and robust benchmarks.

## 5 CONCLUSION

In this work, we provide a critical analysis of RL-based reasoning benchmarks, arguing that high scores are often a deceptive result of mastering brittle, benchmark-specific skills rather than true generalization. Our empirical investigation, using novel diagnostics like the OPG, reveals severe benchmark flaws such as high homogeneity and data redundancy. Subsequent stress tests confirm that the learned skills are remarkably fragile, evidenced by a stark asymmetric generalization across difficulty levels, and failures on out-of-distribution and counterfactual tasks. The primary contribution is not merely this critique but a constructive path forward: we distill our findings into a set of actionable principles—namely difficulty stratification, distributional robustness, and counterfactual reasoning—for designing more reliable, next-generation benchmarks. Adopting these principles is crucial for ensuring that progress is genuine and that we are developing models that are not only capable but also robust and trustworthy. While our experiments focused on specific models and algorithms, this provides a clear direction for future work, which should focus on building these benchmarks and exploring alternative training paradigms to mitigate the identified issues.

## ETHICS STATEMENT

This research adheres to the ICLR Code of Ethics and aims to improve the scientific rigor of benchmark evaluation, in order to drive the development of more robust and trustworthy RL-trained models. The experiments are based on publicly available academic datasets and an in-house dataset compiled from public sources, all used solely for their intended academic analytical purposes. To ensure research integrity, we explicitly state that LLMs were used as tools for specific sub-tasks like data annotation and generation, as well as for language polishing after the manuscript was written. All core scientific contributions, from experimental design to analysis, originate from the human authors. Finally, we recognize that the analyzed models and data may harbor latent societal biases, and while a full audit is beyond the scope of the current work, we consider it an important direction for future research.

## REPRODUCIBILITY STATEMENT

We are committed to ensuring the reproducibility of this research. All experiments are based on publicly available models (the Qwen2.5 series) and datasets (MATH, GSM8K, HeadQA), with full details on hyperparameters and software provided in Appendix A. The core methodology of our research, including the formal definition of the Oracle Performance Gap (OPG) and the design of our three stress tests (Difficulty, Distribution, and Counterfactual), is thoroughly elaborated in Section 2 and Section 3. The specific protocols for our novel data construction methods, such as automated difficulty annotation and counterfactual generation, are detailed in the Appendix (B.1, D, E.1). Furthermore, detailed data tables and qualitative case studies are also available in the Appendix. We believe these detailed descriptions are sufficient to support the reproduction of this work. All source code will be made available upon acceptance of the manuscript.

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

APPENDIX

# A    FULL EVALUATION SETUP

## A.1    POST-TRAINING METHODS

**Reinforcement Learning** Reinforcement Learning (RL) has recently proven effective at steering large language models toward complex, multi-step objectives by optimizing policies with scalar reward signals (Zeng et al., 2025). For our experiments, we utilized the `easy-r1` framework, a fork of the original `veRL` project (Zheng et al., 2025). We employed its implementation of the Group Relative Policy Optimization (GRPO) (Shao et al., 2024) algorithm to fine-tune the `Qwen2.5-7B-Instruct` model, using final answer correctness as the reward signal. Our RL configuration uses a learning rate of $1 \times 10^{-6}$ with the AdamW optimizer and a weight decay of $1.0 \times 10^{-2}$. We generate 5 responses per prompt with a maximum total sequence length of 4096 tokens, using a temperature of 1.0 and a top-p of 0.99. The model is updated with a global batch size of 16. KL-divergence regularization was enabled with a coefficient of $1.0 \times 10^{-2}$. We trained the model for 5 epochs and selected the checkpoint with the best validation performance.

**Supervised Fine-Tuning** Supervised Fine-Tuning (SFT) remains a fundamental technique for adapting large pre-trained models by directly minimizing cross-entropy on high-quality datasets (Parashar et al., 2025). We use the LLaMA-Factory framework (Zheng et al., 2024), which is an extensible and user-friendly framework supporting multiple architectures and advanced optimization algorithms, to fine-tune our model on teacher-generated chain-of-thought traces. We use $1 \times 10^{-6}$ as learning rate, the batch size is 512 and we train for 5 epoch to align with our RL settings.

## A.2    DATASETS AND BENCHMARKS

Our analysis was conducted across the following four benchmarks, chosen to cover a range of mathematical and general reasoning tasks:

- **MATH** (Hendrycks et al., 2021): A challenging dataset of 12,500 competition mathematics problems designed to test mathematical problem-solving.

- **GSM8K** (Cobbe et al., 2021): A dataset of 8,500 high-quality, linguistically diverse grade school math word problems created to measure multi-step reasoning.

- **HeadQA** (Vilares & Gómez-Rodríguez, 2019): A multiple-choice question answering dataset sourced from Spanish medical board exams, covering a wide range of topics and requiring specialized knowledge.

- **DeepScaler** (Luo et al.): A proprietary, in-house dataset created to evaluate specific mathematical reasoning abilities. It contains approximately 40,000 unique math problem-answer pairs compiled from sources like the AIME, AMC, Omni-MATH, and Still datasets.

## A.3    IMPLEMENTATION DETAILS

All experiments were conducted on a single server equipped with 4 NVIDIA A100 (80GB) GPUs. Our implementation relies on PyTorch and the Hugging Face Transformers library.

# B    DETAILED DATA FOR DIFFICULTY-STRATIFIED ANALYSIS

## B.1    AUTOMATED DIFFICULTY LEVEL ANNOTATION

To ensure a systematic and reproducible partitioning of our datasets into difficulty levels (L1-L5), we employed an automated annotation pipeline. Instead of relying on subjective manual labeling, we developed a detailed rubric based on the cognitive complexity required for each problem and used a large language model (`Gemini 2.5 Pro`) to assign a difficulty score to each problem in our corpus.

The process was guided by the five-level standard defined below. For each problem, the full text of this rubric was provided to the LLM, which was then prompted to return the single most appropriate difficulty level.

**Level 1: Direct Application of Basic Rules.** Problems that can be solved in one or two steps, where each step is a direct application of a basic formula or operational rule. The solution path is linear and requires minimal strategic planning.

**Level 2: Identification of Standard Models.** Problems that require identifying the correct standard model or general formula from a set of known methods. This tests for "pattern recognition" of classic problem types.

**Level 3: Multi-Step, Cross-Conceptual Planning.** Problems that cannot be solved by a single standard model and require a coherent plan that links multiple concepts or steps, often from different mathematical areas.

**Level 4: Application of Abstract Concepts.** Problems requiring a deep understanding and flexible application of a major, abstract mathematical theory. The solution process is often non-intuitive and relies on a foundational result within a branch of mathematics.

**Level 5: Axiomatic Reasoning and Creation.** Problems that require reasoning "from first principles" within an axiomatic framework. This involves performing logical deductions, constructing proofs, or finding counterexamples based on the foundational rules of a mathematical structure.

The entire dataset was processed using a parallelized script with a thread pool executor to efficiently query the LLM API. The script included robust error handling and checkpointing to ensure the complete and accurate annotation of the corpus.

## B.2 RESULT

This section provides the full cross-difficulty generalization performance matrices that form the basis for the analysis in Section 3.1.1 and the visualizations in Figure 2. Table 5 presents the results for the `Qwen2.5-3B-Instruct` model, and Table 6 presents the results for the `Qwen2.5-7B-Instruct` model.

The data in these tables highlights the two key phenomena discussed in the main text. First, the asymmetric generalization is visible by comparing the top-right and bottom-left quadrants of the matrices. For instance, in Table 6, the model trained on Level 5 achieves 94.50% on Level 1, while the model trained on Level 1 only achieves 52.00% on Level 5. Second, the deceptive nature of the average score is evident in the rightmost 'Average' column, where the scores for all five specialist models are remarkably similar (e.g., ranging only from 78.60% to 80.10% for the 7B model), despite their vastly different generalization profiles.

Table 5: Cross-Difficulty Generalization Performance Matrix for the *Qwen2.5-3B-Instruct* model. All values are pass@1 accuracy.

| Trained on | Evaluated on Training Set of Level | | | | | |
|---|---|---|---|---|---|---|
| | Level 1 | Level 2 | Level 3 | Level 4 | Level 5 | Average |
| **Level 1** | 94.50% | 85.00% | 71.00% | 66.00% | 41.00% | 71.50% |
| **Level 2** | 93.00% | 87.50% | 73.00% | 65.00% | 42.50% | 72.20% |
| **Level 3** | 92.50% | 86.00% | 75.00% | 66.00% | 40.00% | 71.90% |
| **Level 4** | 92.50% | 86.50% | 72.00% | 68.00% | 43.00% | 72.40% |
| **Level 5** | 94.00% | 87.00% | 73.00% | 62.00% | 46.50% | 72.50% |
| **Original** | 92.00% | 83.50% | 69.50% | 62.50% | 43.50% | 70.20% |

Table 6: Cross-Difficulty Generalization Performance Matrix for the *Qwen2.5-7B-Instruct* model. All values are pass@1 accuracy.

| Trained on | Evaluated on Training Set of Level | | | | | |
|---|---|---|---|---|---|---|
| | Level 1 | Level 2 | Level 3 | Level 4 | Level 5 | Average |
| Level 1 | 97.00% | 90.00% | 78.00% | 76.00% | 52.00% | 78.60% |
| Level 2 | 94.00% | 91.50% | 82.50% | 76.00% | 54.00% | 79.60% |
| Level 3 | 95.50% | 91.00% | 83.50% | 72.50% | 56.50% | 79.80% |
| Level 4 | 93.50% | 88.50% | 81.00% | 80.00% | 57.00% | 80.00% |
| Level 5 | 94.50% | 91.00% | 78.00% | 73.00% | 64.00% | 80.10% |
| Original | 95.50% | 87.50% | 76.50% | 74.00% | 52.00% | 77.60% |

## C   A SUPPLEMENTARY EXPERIMENT TO THE DIFFICULTY TEST

This appendix provides the full performance data for the "generalist-optimized" models described in our supplementary experiment on the difficulty test. The performance lift curves presented in Figure 4 in the main text are directly derived from the raw accuracy scores presented here. Table 7 details the results for the 7B model, while Table 8 shows the results for the 3B model.

**Setup.** To investigate the impact of training data difficulty on final generalization, we conduct a complexity test. We first train five generalist-optimized models, $M_{L_i}$ for $i \in \{1, \ldots, 5\}$, on the previously defined difficulty-stratified training sets, $\mathcal{D}_{\text{train}}^{L_i}$. The key difference from our prior analysis lies in the evaluation protocol, which is centered around a novel, balanced test set.

- **Test_Balanced:** This is the unified and balanced evaluation suite, constructed by sampling an equal number of problems from each of the five difficulty levels. This results in a test set $\mathcal{D}_{bal}$ composed of five equal-sized partitions, $\{\mathcal{D}_{\text{test, bal}}^{L_j}\}_{j=1}^{5}$.

Unlike the models in the first experiment, these models are "generalist-optimized," meaning we select the checkpoint for each $M_{L_i}$ with the highest overall accuracy on the Test_Balanced set.

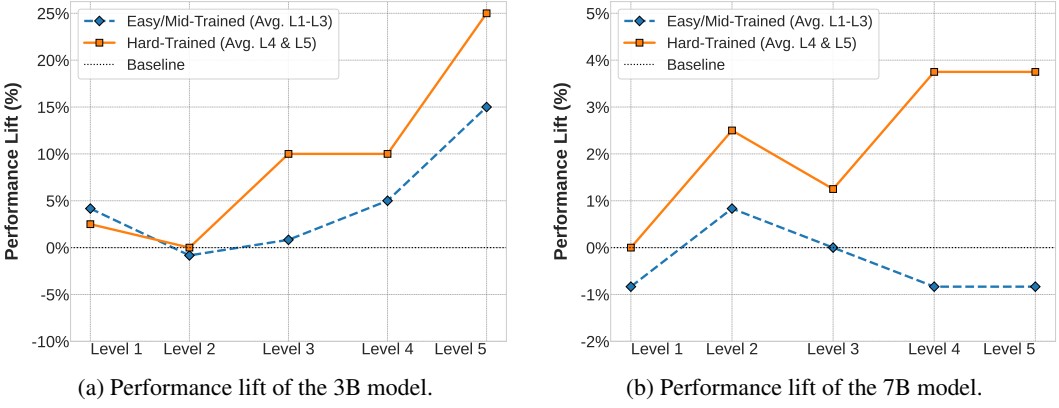

(a) Performance lift of the 3B model.    (b) Performance lift of the 7B model.

Figure 4: *Asymmetric Generalization is consistent across model scales.* Across both the 3B model (a) and the 7B model (b), training on high-difficulty problems (L4-L5, orange line) yields a uniformly superior performance lift over training on easier problems (L1-L3, blue line), proving that mastering complexity is essential for acquiring robust, transferable skills. Full performance data is provided in Table 8 and Table 7.

Our complexity test reveals a stark pattern of asymmetric generalization, as illustrated in Figure 4. Models trained on high-difficulty problems (L4-L5) demonstrate a uniformly superior performance profile, outperforming their counterparts trained on easier data (L1-L3) across all evaluated task complexities. This finding has a critical implication for how we create datasets to train capable models: **the training data must include a significant proportion of difficult problems**. Therefore, for benchmark suites to drive meaningful progress, it is crucial that their provided training sets are sufficiently challenging to promote the development of truly robust models. The data in these

tables clearly illustrates the "asymmetric generalization" phenomenon. For example, in Table 8, the model trained on Level 1 ($M_{L_1}$) achieves high accuracy (97.50%) on Level 1 test problems but sees its performance drop to just 32.50% on Level 5 problems. In contrast, the model trained on Level 5 ($M_{L_5}$) maintains robust performance across all levels, demonstrating a more generalizable capability.

Table 7: Performance of *Qwen2.5-7B* generalist-optimized models on the balanced test set. Each row represents a model trained on a specific difficulty level ($L_i$), evaluated across test questions of all five difficulty levels.

| Trained on | Evaluated on Test Set Questions of Level | | | | | |
|---|---|---|---|---|---|---|
| | Level 1 | Level 2 | Level 3 | Level 4 | Level 5 | Average |
| **Level 1** | 97.50% | 90.00% | 82.50% | 75.00% | 50.00% | 79.00% |
| **Level 2** | 95.00% | 90.00% | 80.00% | 77.50% | 47.50% | 79.00% |
| **Level 3** | 97.50% | 85.00% | 85.00% | 77.50% | 50.00% | 79.00% |
| **Level 4** | 97.50% | 87.50% | 85.00% | 80.00% | 55.00% | 81.00% |
| **Level 5** | 97.50% | 92.50% | 82.50% | 82.50% | 52.50% | 81.50% |
| **Original** | 97.50% | 87.50% | 82.50% | 77.50% | 50.00% | 79.00% |

Table 8: Performance of *Qwen2.5-3B* generalist-optimized models on the balanced test set. The performance decay for models trained on easy levels (L1, L2) is particularly pronounced.

| Trained on | Evaluated on Test Set Questions of Level | | | | | |
|---|---|---|---|---|---|---|
| | Level 1 | Level 2 | Level 3 | Level 4 | Level 5 | Average |
| **Level 1** | 97.50% | 82.50% | 75.00% | 72.50% | 32.50% | 72.00% |
| **Level 2** | 95.00% | 87.50% | 80.00% | 65.00% | 35.00% | 72.00% |
| **Level 3** | 97.50% | 90.00% | 80.00% | 72.50% | 45.00% | 77.00% |
| **Level 4** | 95.00% | 87.50% | 87.50% | 75.00% | 47.50% | 78.50% |
| **Level 5** | 95.00% | 87.50% | 87.50% | 75.00% | 47.50% | 78.50% |
| **Original** | 92.50% | 87.50% | 77.50% | 65.00% | 22.50% | 69.00% |

# D    DATA CONSTRUCTION PROTOCOL FOR THE DISTRIBUTION TEST

This section details the step-by-step procedure used to construct the specialized training and test sets for the Distribution Test, as described in Section 3.2.1. The entire process is designed to create a controlled environment for measuring generalization as a function of semantic distance. The process consists of three main stages:

**Step 1: Semantic Embedding and Clustering.**    We began with our full corpus of approximately 44785 mathematics problems. To understand their semantic relationships, we first encoded each problem into a high-dimensional vector representation using the `all-mpnet-base-v2` sentence encoder. We then applied K-Means clustering to this high-dimensional embedding space. Using a combination of the Elbow method and Silhouette score analysis, we determined the optimal number of clusters to be $k = 3$, effectively partitioning the entire dataset into three broad, semantically coherent groups.

**Step 2: Core Training Set (`Train_Core`) Selection.**    Our goal was to create a highly concentrated, semantically narrow training set. To achieve this, we first projected the high-dimensional embeddings into a 2D space using t-SNE for visualization. We then focused on a single target cluster (e.g., Cluster 1). Instead of sampling from the high-dimensional space, our selection was based on the *visual density* in the 2D projection. Using the 'NearestNeighbors' algorithm on the 2D t-SNE coordinates, we identified the point within the target cluster whose 2,000 nearest neighbors occupied the smallest possible Euclidean radius. These 2,000 points, representing the most visually compact region of the cluster, formed our exclusive `Train_Core` training set.

**Step 3: Distance-Stratified Test Set Construction.** To create test sets with increasing semantic distance, we used the remaining 42785 problems not selected for `Train_Core`. First, we calculated the geometric centroid of the 2,000 `Train_Core` points in the 2D t-SNE space. Then, for every other point in the dataset, we computed its Euclidean distance to this centroid. All candidate test points were then sorted based on this distance, from nearest to farthest. This sorted list was partitioned into five equal-sized bins. Finally, we randomly sampled 80 problems from each bin to create our five final test sets, D1 (semantically closest) through D5 (semantically farthest).

The entire data construction pipeline is visually summarized in Figure 5. Panel (a) illustrates the outcome of the `Train_Core` selection process described in Step 2, while Panel (b) shows the resulting distribution of the five distance-stratified test sets as detailed in Step 3.

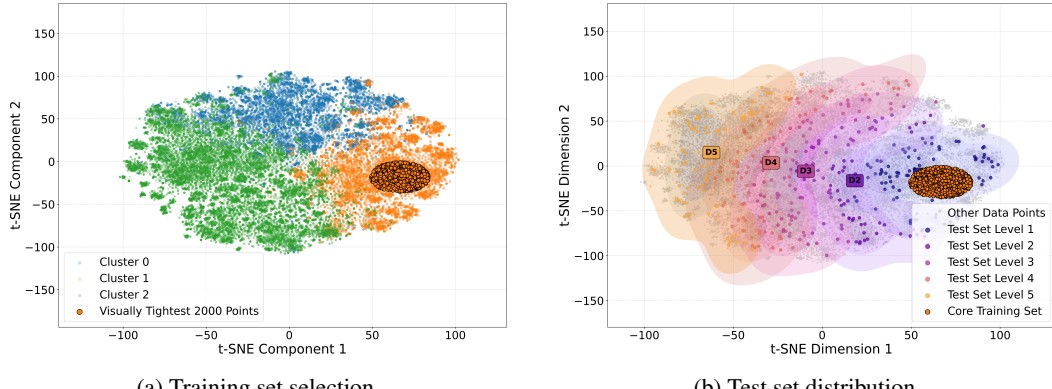

(a) Training set selection.   (b) Test set distribution.

Figure 5: *Visualization of the experimental data construction for the distribution test.* (a) The highly concentrated $\mathcal{D}_{\text{core}}$ set is selected from a semantic cluster. (b) The test sets are sampled and binned based on their increasing semantic distance from the $\mathcal{D}_{\text{core}}$ centroid.

# E   THE COUNTERFACTUAL ROBUSTNESS TEST

This section provides detailed, qualitative examples of how fine-tuned models fail on counterfactual reasoning tasks, as discussed in Section 3.2.2. Each table analyzes a specific failure case, comparing the required reasoning path (based on the novel, counterfactual premise) with the model's actual thought process. These examples concretely illustrate the models' strong tendency to disregard explicit instructions and default to their pre-trained, memorized knowledge.

## E.1   METHODOLOGY: AUTOMATED DATASET GENERATION

To ensure the diversity and systematic nature of our counterfactual examples, we developed and executed the following automated pipeline, moving beyond manual creation.

**Step 1: Strategy — LLM as Data Creator.** Our core strategy was to leverage a powerful Large Language Model to act as a creative research assistant. This approach allows for the large-scale and consistent application of complex transformation rules needed to create a high-quality counterfactual dataset.

**Step 2:Task Definition — The Counterfactual Transformation.** We provided the LLM (`Gemini 2.5 Pro`) with a detailed, multi-step prompt that precisely defined the transformation task. The instructions guided the model to first analyze a given standard problem to identify a core logical or mathematical rule. Subsequently, the model was tasked to invent a plausible but contrary-to-fact rule, rewrite the problem statement to include this new premise, and finally, generate a new step-by-step solution based exclusively on the novel rule.

**Step 3: Execution — Parallelized Pipeline.** This generation process was applied to our entire source dataset. To manage the scale, the pipeline was executed in parallel using a Python script with a `ThreadPoolExecutor` to handle concurrent API requests. The full, unabridged master prompt used in this process is available in our supplementary materials to ensure full reproducibility.

## E.2    CASE STUDY: ARITHMETIC ORDER OF OPERATIONS

**(Counterfactual Premise)**

A novel order of operations, **PESAMD**, is defined: Parentheses, Exponents, **S/A**, then **M/D**. The model is asked to evaluate $f(x) = \frac{3x-2}{x-2}$.

**Correct Reasoning (PESAMD)**
1. **Numerator (S first):** $3 \times (0-2) = -6$
2. **Denominator:** $0 - 2 = -2$
3. **Division (last):** $\frac{-6}{-2} = 3$

The final correct answer is **9**.

**Model's Actual Reasoning**
1. **Numerator (M first):** It computes $3 \times 0 = 0$ first, then $0 - 2 = -2$. This follows the **memorized PEMDAS rule**, violating the premise.
2. **Denominator:** Correctly computes $0 - 2 = -2$.
3. **Division:** $\frac{-2}{-2} = 1$.

The final incorrect answer is $\frac{14}{3}$.

## E.3    CASE STUDY: NUMBER THEORY DIVISOR RULE

**(Counterfactual Premise)**

A new system defines the number of divisors of $N = p_1^{a_1} \cdots$ as the **sum** of $(a_i + 1)$ values. Find the number of divisors for $N = 12$.

**Correct Reasoning (Sum Rule)**
1. Prime factorization of 12 is $2^2 \times 3^1$.
2. The exponents are $a_1 = 2, a_2 = 1$.
3. Apply the new **sum rule**: $(2+1)+(1+1) = 5$.

The final correct answer is **5**.

**Model's Actual Reasoning**
1. Correctly finds prime factorization: $12 = 2^2 \times 3^1$.
2. Ignores the "sum" rule and applies the memorized "product" rule: $(2 + 1) \times (1 + 1) = 6$.

The final incorrect answer is **6**.

## E.4    CASE STUDY: PHYSICS SPEED FORMULA

**(Counterfactual Premise)**

A car travels 120 km in 2 hours. In this reality, 'average speed' is calculated as: **speed = time / distance**. Find the speed.

**Correct Reasoning (New Formula)**
1. Identify Time = 2 hours, Distance = 120 km.
2. Apply the new formula time / distance: $2 \div 120 = \frac{1}{60}$.

The final correct answer is $\frac{1}{60}$ **km/h**.

**Model's Actual Reasoning**
1. Correctly identifies Time and Distance.
2. Ignores the new formula and applies the memorized, standard formula 'distance / time': $120 \div 2 = 60$.

The final incorrect answer is **60 km/h**.

# F USAGE OF LARGE LANGUAGE MODELS

We declare that LLMs were used solely for language polishing purposes in this work. Specifically, after completing the initial draft entirely through human effort, we employed LLM assistance exclusively for grammatical refinement and improving the clarity of English expression to meet academic writing standards. All intellectual contributions, from conceptualization to initial manuscript preparation, were performed by the human authors. The use of LLM was limited to post-writing language enhancement, similar to traditional proofreading services, ensuring that non-native English speakers can present their research with appropriate linguistic quality while maintaining complete authorship and originality of the scientific content.

