# OpenReview forum: "Rethinking RL Evaluation: Can Benchmarks Truly Reveal Failures of RL Methods?"
_ICLR.cc/2026/Conference — ICLR 2026 Conference Withdrawn Submission_

### Official Review · Reviewer_8qqd · 2025-10-18

**Soundness:** 2
**Presentation:** 3
**Contribution:** 2
**Rating:** 4
**Confidence:** 3

**Summary:**

This paper considers the problem of designing benchmarks for reinforcement learning post-training of LLMs. Using two base models from the Qwen family, it shows experimentally on four datasets that the performance is similar when trained on the training set and test set (characterized by a proposed metric called Oracle Performance Gap), suggesting that current experimental protocols are not sufficient. With more fine-grained experiments, the paper shows that 1) performance should be stratified by task difficulty as average metrics can be misleading, 2) performance on out-of-distribution tasks can become worse after post-training and thus should be reported, and 3) benchmarks should test for counterfactual reasoning, as models can default to reciting memorized knowledge.

**Strengths:**

Overall, the paper is well-written and provides the reader with motivation and recommends a clear course of action. Both the text and experiments are organized clearly and the results are presented with the hypothesis/conclusion paradigm that made the paper easy to understand. The design of the experiments seem to be sound.

**Weaknesses:**

To make the experimental results more strongly support the conclusions, standard errors should be included, as they are critical for rigorous statistical testing of hypotheses.

Another weakness is that the conclusions provided in this paper have already been known elsewhere in the machine learning literature to some extent. The fact that the average performance metric may be misleading is Simpson's Paradox [1], leading to the development of metrics like precision and recall for classification. The importance of testing out-of-distribution generalization is known in reinforcement learning [2]. There is a benchmark for testing LLM's ability for causal inference, which is important in real-world applications [3]. This paper shows that these same phenomena occur in RL-based post-training for LLMs, which is interesting but not really unexpected.

[1] https://en.wikipedia.org/wiki/Simpson%27s_paradox
[2] Packer et al. "Assessing Generalization in Deep Reinforcement Learning", arXiv preprint arXiv:1810.12282.
[3] Du et al. "Ice Cream Doesn’t Cause Drowning: Benchmarking LLMs Against Statistical Pitfalls in Causal Inference", arXiv preprint arXiv:2505.13770.

**Questions:**

- How were the hyperparameters selected?
- Is there a way to interpret the semantic clusters described in section 3.2.1? Do they correspond to certain topics?

---

> ### Author Response · Authors · 2025-12-03
>
> ### Response to Reviewer
>
> We thank the reviewer for the insightful comments, particularly for highlighting the connection to Simpson's Paradox and emphasizing the need for statistical rigor. Below we address your concerns with new experimental data and clarifications.
>
> ### Question 1:
> >"To make the experimental results more strongly support the conclusions, standard errors should be included, as they are critical for rigorous statistical testing of hypotheses."
>
> ### Response:
> We agree that statistical rigor is vital. While retraining LLMs multiple times is computationally prohibitive, we have rigorously quantified the stability of our model's performance by conducting 10 independent evaluation runs using sampling (Temperature > 0) for each reported metric. We also updated Table 1 in Section 2.2 to report the Mean $\pm$ 95% CI over 10 runs.
>
> The table below reports the Mean Accuracy $\pm$ 95% Confidence Interval (CI). The use of 95% CI ($N=10$) provides a stricter bound than standard errors.
>
> **Table 1: Performance Stability (Mean $\pm$ 95% CI, N=10 Runs)**
>
> | Benchmark | Size | 10% Train | 20% Train | 50% Train | 100% Train | Test (Oracle) |
> | :---: | :---: | :---: | :---: | :---: | :---: | :---: |
> | MATH | 3B | 63.88% $\pm$ 1.04 | 65.18% $\pm$ 0.84 | 64.84% $\pm$ 1.25 | 64.62% $\pm$ 0.98 | 64.62% $\pm$ 1.11 |
> | MATH | 7B | 73.64% $\pm$ 0.48 | 73.28% $\pm$ 0.68 | 73.04% $\pm$ 0.91 | 74.04% $\pm$ 0.39 | 74.00% $\pm$ 0.68 |
> | GSM8K | 3B | 82.95% $\pm$ 0.38 | 83.93% $\pm$ 0.47 | 86.93% $\pm$ 0.34 | 87.04% $\pm$ 0.49 | 87.98% $\pm$ 0.35 |
> | GSM8K | 7B | 88.76% $\pm$ 0.47 | 89.58% $\pm$ 0.44 | 91.14% $\pm$ 0.30 | 91.72% $\pm$ 0.31 | 91.87% $\pm$ 0.31 |
> | DeepScaler | 7B | 42.05% $\pm$ 0.84 | 42.09% $\pm$ 0.57 | 42.84% $\pm$ 0.73 | 42.36% $\pm$ 0.77 | 42.64% $\pm$ 0.75 |
> | HeadQA | 3B | 62.98% $\pm$ 0.77 | 65.90% $\pm$ 1.19 | 67.16% $\pm$ 0.79 | 67.24% $\pm$ 0.59 | 67.57% $\pm$ 0.70 |
> | HeadQA | 7B | 72.94% $\pm$ 0.90 | 72.90% $\pm$ 0.79 | 74.39% $\pm$ 0.60 | 75.20% $\pm$ 0.66 | 75.60% $\pm$ 0.50 |

---

> > ### Author Response · Authors · 2025-12-03
> >
> > ### Question 2:
> > >"Another weakness is that the conclusions provided in this paper have already been known elsewhere in the machine learning literature to some extent. The fact that the average performance metric may be misleading is Simpson's Paradox [1], leading to the development of metrics like precision and recall for classification. The importance of testing out-of-distribution generalization is known in reinforcement learning [2]. There is a benchmark for testing LLM's ability for causal inference, which is important in real-world applications [3]. This paper shows that these same phenomena occur in RL-based post-training for LLMs, which is interesting but not really unexpected."
> >
> > ### Response:
> > While these concepts are theoretically known, our contribution is providing the first systematic empirical evidence that they are actively compromising current RL-LLM evaluations. We discover and bring up that this issue also occurs in RL-LLM evaluation. We demonstrate that SOTA scores on leaderboards like GSM8K are often illusory—achieved via "benchmark overfitting" (Low OPG) rather than genuine reasoning. Furthermore, we propose the OPG metric and specific Stress Tests as a concrete diagnostic framework to detect these failures, filling a critical gap in standard LLM evaluation protocols.

---

> > > ### Author Response · Authors · 2025-12-03
> > >
> > > ### Question 3:
> > > >"How were the hyperparameters selected?"
> > >
> > > ### Response:
> > > To ensure reproducibility and representativeness, we strictly followed the default configuration of the EasyR1 framework, a widely adopted open-source RL codebase. We did not perform extensive hyperparameter tuning to "game" the metrics; instead, we used these community standards to reflect the typical performance practitioners would observe. Full details are provided in Appendix A.

---

> > > > ### Author Response · Authors · 2025-12-03
> > > >
> > > > ### Question 4:
> > > > >"Is there a way to interpret the semantic clusters described in section 3.2.1? Do they correspond to certain topics?"
> > > >
> > > > ### Response:
> > > > Yes. By examining specific problem instances within each cluster, we identified a clear semantic progression, which explains the 'Performance Inversion' we observed.
> > > >
> > > > * **Training Core ( $d_0$ ):** The core cluster selected for training consists predominantly of Geometry problems (e.g., calculating areas of triangles, properties of circles, and spatial diagrams).
> > > > * **Semantic Shift ($d_1 \rightarrow d_5$):** As the semantic distance increases, the test sets gradually shift away from this geometric domain:
> > > >     * **Near-Distribution ($d_1 - d_2$):** Still contains some geometry but introduces standard Algebra and Number Theory.
> > > >     * **Far-Distribution ($d_4 - d_5$):** The problems shift completely to Abstract Logic, Combinatorics, and Non-standard Puzzles (e.g., sequence pattern recognition, permutation inequalities) that share almost no structural similarity with the geometry-focused training set.
> > > >
> > > > This confirms that our Distribution Test effectively measures the model's inability to transfer reasoning skills from a specific domain (Geometry) to semantically distinct domains (Logic/Combinatorics), resulting in the observed performance collapse.

---

### Official Review · Reviewer_9nw4 · 2025-10-28

**Soundness:** 2
**Presentation:** 1
**Contribution:** 2
**Rating:** 2
**Confidence:** 3

**Summary:**

The paper aims to improve the scientific rigour of benchmarks, primarily by measuring how well RL-finetuning of language models generalises out of distribution. It introduces three methods for creating distributional shifts: splitting up existing datasets labeled by difficulty, clustering data points by their vector representations and creating new datasets by e.g. introducing new axioms in maths. The paper also makes a number of recommendations for how future benchmarks should be constructed.

**Strengths:**

The paper demonstrates a few interesting patterns:

**SFT shows less performant train-test generalisation than RL-finetuning**
SFTed models show substantial test-set-performance differences when trained on training vs test splits (MATH: 17-24% on test when trained on train set; 40-64% when trained on test set and GSM8K: 17-20% vs 68-79%). In contrast,  RL finetuned models do not show any substantial difference (MATH: 64.2% vs 64.4% (3B) and 74.8% vs 74.4% (7B); GSM8K: 87.5% vs 88.9% (3B) and 91.1% vs 91.8% (7B)).

**RL finetuned models can fail to generalise between semantically cluster data subsets**
When datasets are split using semantic clustering rather than I.I.D, RL-fine tuning on one cluster can actually slightly decrease performance on another (although this effect size is small and  its unclear if these results are statistically significant).

I believe some of the contributions of this paper, for example the semantic clustering methodology, are somewhat interesting and novel.

**Weaknesses:**

Core to the contributions of this paper are new results, utilising new methods of analysis on existing benchmarks. However, I am uncertain about the novelty and clarity of the communication of these results; and am particularly concerned about misleading presentation. The experiments also lack statistical rigour.

### “Novel Diagnostic Framework.”
The paper claims a core contribution of the paper is its “Novel Diagnostic Framework”. However, I find the aggregate statistics that are introduced to be not particularly novel, and potentially misleading. Presenting the raw data would arguably have been less confusing and less likely to mislead readers.

#### I think the 'Average Cross-Difficulty Generalization' metric may be misleading.
If I am understanding Figure 3 correctly, the 'Average Cross-Difficulty Generalization' metric excludes each model's own training level from the average. This means the L1-trained model is evaluated on the average of Levels 2-5 (mostly harder problems), while the L5-trained model is evaluated on the average of Levels 1-4 (mostly easier problems). This creates an inherent bias in the comparison - the L5 model appears to 'generalize better' simply because it's being tested on an easier set of problems. The monotonic pattern highlighted in the paper would (as far as I can tell) still appear if the same model was used across all data points. Please could the authors clarify whether I have misunderstood this figure?

#### The “Oracle Performance Gap” is similar to
Similarly, the “Oracle Performance Gap” is not particularly helpful relative to just reporting raw numbers on train/test set. Further, I’m not clear on why the “oracle performance gap” is actually different to just reporting the difference between train/test performance: they in fact fine tune on the “test set” and report performance on the test set, but this is effectively just a new train set.

### Missing statistical significance
The paper presents empirical findings without adequate statistical analysis. There are no confidence intervals, significance tests, or multiple seeds reported for key experiments. Statistical tests are essential for a paper aiming to improve evaluation rigour.

### Additional concerns:
#### Normative framing.
Rather than presenting empirical observations and letting readers evaluate their implications, the authors make prescriptive claims about benchmark design. They introduce "principles" and assert that benchmarks "must" include certain features, without establishing clear criteria for what constitutes good generalization or why their specific recommendations follow from their findings.
I think Principle 1 does this relatively well: the paper says evaluations “must be balanced and stratified” in order to prevent “masking”. However, it essentially restates the well-established practice of disaggregated reporting. Principle 2 is more problematic, declaring that "faithful" benchmarks must test distributional robustness without justifying why this is necessary. Of course, in many applications, robustness to distributional shifts is important. However, a simple normative claim that distributional robustness should be measured does not add value to the paper.

#### Counterfactual test
Finally, the counterfactual test reveals an interesting limitation of LLMs but doesn't specifically relate to RL methods or benchmark design.

**Questions:**

* Have I understood Figure 3 properly? Does the trend persist if you control for the increased difficulty of test sets used for L1 vs L5?
* Are multiple seeds used for any of the experiments?

---

> ### Author Response · Authors · 2025-12-03
>
> ### Response to Reviewer
>
> We thank the reviewer for the comments. Regarding the questions on the difficulty metric and the normative framing, we provide the following clarifications to address the concerns.
> ### Question 1:
> > "If I am understanding Figure 3 correctly, the 'Average Cross-Difficulty Generalization' metric excludes each model's own training level from the average. This means the L1-trained model is evaluated on the average of Levels 2-5 (mostly harder problems), while the L5-trained model is evaluated on the average of Levels 1-4 (mostly easier problems). This creates an inherent bias in the comparison - the L5 model appears to 'generalize better' simply because it's being tested on an easier set of problems. The monotonic pattern highlighted in the paper would (as far as I can tell) still appear if the same model was used across all data points. Please could the authors clarify whether I have misunderstood this figure?"
>
> ### Response:
> Regarding the concern that the metric definition in Figure 3 might introduce a difficulty shift, we emphasize that this factor was explicitly controlled for in our Supplementary Experiment (Appendix C). Our results confirm that the observed trend is robust and not an artifact of the metric definition.
>
> **Evidence from Balanced Test Set (Figure 4 & Tables 7-8)**
>
> In Appendix C, we evaluated all specialist models on a unified Balanced Test Set ($\mathcal{D}_{bal}$) containing an equal number of problems from all levels (L1-L5). This ensures every model is evaluated on the exact same distribution.
>
> **Table 1:** Qwen2.5-7B on Balanced Test Set
>
> | Model Training Source | Average Accuracy on $\mathcal{D}_{bal}$ |
> | :---: | :---: |
> | Trained on Level 1 (Easy) | 79.00% |
> | Trained on Level 5 (Hard) | 81.50% |
>
> Furthermore, looking at the breakdown (Table 8):
>
> * Transfer to Hard: The L5-trained model achieves 47.50% on L5 test data, whereas the L1-trained model collapses to 32.50%.
> * Transfer to Easy: The L5-trained model retains 97.50% accuracy on L1 test data(same as  L1-trained model).
>
> **Conclusion**
>
> The findings in Figure 4 (Balanced Test) confirm the pattern in Figure 3: Training on harder problems yields a strictly superior generalist policy.

---

> ### Author Response · Authors · 2025-12-03
>
> ### Question 2:
> > "Similarly, the “Oracle Performance Gap” is not particularly helpful relative to just reporting raw numbers on train/test set. Further, I’m not clear on why the “oracle performance gap” is actually different to just reporting the difference between train/test performance: they in fact fine tune on the “test set” and report performance on the test set, but this is effectively just a new train set."
>
> ### Response:
> We acknowledge the reviewer's observation that the Oracle model treats the test set as training data. However, this is precisely the logic underpinning the metric. We introduce OPG not to report performance, but as a diagnostic tool to measure benchmark validity.We have adjusted the framing of this part in Section 2.1.1 (Oracle Performance Gap) of the revision.
>
> **Why Raw Numbers are Insufficient**
>
> Reporting raw accuracy (e.g., "Model A achieves 90%") is ambiguous. We do not know if 90% represents true reasoning mastery or simply that the dataset is easy to fit. The Oracle model ($M_{test}$) establishes the Empirical Upper Bound—the maximum possible performance achievable by this architecture when it is allowed to fully memorize the test distribution.
>
> **Interpreting the OPG Values**
>
> * **If OPG $\approx 0$ (Saturation):** This signals that the information content of the training set is functionally identical to the test set. The model gains no advantage even when allowed to "peek" at the test answers (Oracle training), proving the test set lacks distinctive variance.
> * **If OPG $< 0$ (Benchmark Failure):** This is a critical pathological case observed in our experiments (e.g., DeepScaler). It implies that training on the robust training set yields better results than "cheating" on the test set.
>
> **Conclusion**
>
> If a "leaked" model cannot even match the standard model, the test set has fundamentally failed as a reliable evaluation standard. OPG reveals structural insights—saturation ($\approx 0$) or statistical invalidity ($< 0$)—that raw accuracy numbers alone cannot detect.

---

> ### Author Response · Authors · 2025-12-03
>
> ### Question 3:
> > "The paper presents empirical findings without adequate statistical analysis. There are no confidence intervals, significance tests, or multiple seeds reported for key experiments. Statistical tests are essential for a paper aiming to improve evaluation rigour."
>
> ### Response:
> We agree that statistical rigor is vital. While retraining LLMs multiple times is computationally prohibitive, we have rigorously quantified the stability of our model's performance by conducting 10 independent evaluation runs using sampling for each reported metric. We also updated Table 1 in Section 2.2 to report the Mean $\pm$ 95% CI over 10 runs.
>
> The table below reports the Mean Accuracy $\pm$ 95% Confidence Interval (CI).
>
> **Table 1: Performance Stability (Mean $\pm$ 95% CI, N=10 Runs)**
>
> | Benchmark | Size | 10% Train | 20% Train | 50% Train | 100% Train | Test (Oracle) |
> | :---: | :---: | :---: | :---: | :---: | :---: | :---: |
> | MATH | 3B | 63.88% $\pm$ 1.04 | 65.18% $\pm$ 0.84 | 64.84% $\pm$ 1.25 | 64.62% $\pm$ 0.98 | 64.62% $\pm$ 1.11 |
> | MATH | 7B | 73.64% $\pm$ 0.48 | 73.28% $\pm$ 0.68 | 73.04% $\pm$ 0.91 | 74.04% $\pm$ 0.39 | 74.00% $\pm$ 0.68 |
> | GSM8K | 3B | 82.95% $\pm$ 0.38 | 83.93% $\pm$ 0.47 | 86.93% $\pm$ 0.34 | 87.04% $\pm$ 0.49 | 87.98% $\pm$ 0.35 |
> | GSM8K | 7B | 88.76% $\pm$ 0.47 | 89.58% $\pm$ 0.44 | 91.14% $\pm$ 0.30 | 91.72% $\pm$ 0.31 | 91.87% $\pm$ 0.31 |
> | DeepScaler | 7B | 42.05% $\pm$ 0.84 | 42.09% $\pm$ 0.57 | 42.84% $\pm$ 0.73 | 42.36% $\pm$ 0.77 | 42.64% $\pm$ 0.75 |
> | HeadQA | 3B | 62.98% $\pm$ 0.77 | 65.90% $\pm$ 1.19 | 67.16% $\pm$ 0.79 | 67.24% $\pm$ 0.59 | 67.57% $\pm$ 0.70 |
> | HeadQA | 7B | 72.94% $\pm$ 0.90 | 72.90% $\pm$ 0.79 | 74.39% $\pm$ 0.60 | 75.20% $\pm$ 0.66 | 75.60% $\pm$ 0.50 |

---

> ### Author Response · Authors · 2025-12-03
>
> ### Question 4:
> > "Rather than presenting empirical observations and letting readers evaluate their implications, the authors make prescriptive claims about benchmark design. They introduce "principles" and assert that benchmarks "must" include certain features, without establishing clear criteria for what constitutes good generalization or why their specific recommendations follow from their findings. I think Principle 1 does this relatively well: the paper says evaluations “must be balanced and stratified” in order to prevent “masking”. However, it essentially restates the well-established practice of disaggregated reporting. Principle 2 is more problematic, declaring that "faithful" benchmarks must test distributional robustness without justifying why this is necessary. Of course, in many applications, robustness to distributional shifts is important. However, a simple normative claim that distributional robustness should be measured does not add value to the paper."
>
> ### Response:
> We fully accept this critique. We recognize that our original language was overly prescriptive. In the final manuscript, we will reframe the "Principles" section to strictly ground our recommendations in our empirical observations, replacing prescriptive terms like "must" with "we recommend" or "our findings suggest." We have also updated Principles 1, 2, and 3 in the revision to align with this approach, replacing prescriptive commands with objective, evidence-based recommendations.

---

### Official Review · Reviewer_b9vn · 2025-10-29

**Soundness:** 2
**Presentation:** 3
**Contribution:** 2
**Rating:** 4
**Confidence:** 4

**Summary:**

This paper challenges the reasonableness of current LLM benchmarks for evaluating RL-based reasoning. The authors find that RL methods achieve near-zero OPG (which is the difference between model trained on test set compared to one trained on train set) and on the other hand, SFT shows large gaps, suggesting that traditional train/test splits may not effectively measure generalization for RL. The paper then proposes 3 tests: difficulty stratification, distribution shift, and counterfactual reasoning.

**Strengths:**

1. This paper covers an important research direction by critically examining benchmark quality and questioning whether current evaluations truly measure the generalization performance.
2. The paper attempts to assess generalization from multiple angles: difficulty, distribution, counterfactual.
3. The writing is generally clear and the paper is easy to follow.
4. The observation that aggregate scores mask cross-difficulty performance differences (Figure 2) is valuable.

**Weaknesses:**

1. A major problem that I have with this submission is that this paper assumes low OPG means "benchmark is broken" but never validates this interpretation. Consider: (a) RL might genuinely learn more robust features than SFT, (b) the test set might be easier than assumed, or (c) benchmark saturation is happening for different reasons than claimed. So for me, here's an alternative interpretation:
(1) RL's low OPG might indicate "good" generalization, not benchmark failure
(2) SFT's high OPG might indicate "poor" generalization, not correct behavior
The authors never tested whether RL actually learned robust policies vs memorized patterns. I would like to propose a simple experiment to validate this: train RL on 10% of train set --> evaluate on the remaining 90% of train set vs. test set. If performance on both is similar, that's genuine generalization. The current setup in this submission can't distinguish this.

2. Table 1: DeepScaler shows OPG = -5.07% (oracle *worse* than train) -> this contradicts the hypothesis entirely, yet the authors don't discuss it

3. Lines 183-186: Authors claim to "rule out data leakage" with one sentence, but provide no contamination analysis of Qwen base models


4. Section 3.2.1's entire Distribution Test uses Euclidean distance in t-sne space as "semantic distance" (in appendix D). I think there is a flaw here as t-SNE only preserves local neighborhood structure, not global distances. Hence I feel that conclusions from Table 3 may be unreliable.

5. The authors used a single llm (gemini 2.5 Pro) to annotate difficulty (in appendix B.1) with zero validation, inter-rater agreement, or comparison to existing labels, like MATH benchmark already has difficulty levels 1-5.

6. [minor]: I feel that this paper has limited scope since most of the analysis was done on math domain. Moreover the authors never experimented with a more challenging benchmark (even if it is from math domain, like FrontierMATH).

**Questions:**

1. How do other RL methods (DPO, PPO, etc.) behave? Is low OPG specific to GRPO or general to offline/online RL? We need to isolate the actual cause i.e. is the algorithm too good that it can easily generalize or is it the benchmark that's quite easy to memorize on?

---

> ### Author Response · Authors · 2025-12-03
>
> ### Response to Reviewer
>
> We thank the reviewer for their insightful feedback. Following your suggestion, we conducted the "10% training split" experiment to directly test the hypothesis regarding generalization versus memorization. Additionally, we re-evaluated the distribution shift using global embeddings and extended our analysis to the DAPO algorithm.The results from this validation align with our original findings and help clarify the nature of the OPG metric. Our detailed responses follow.
>
> ### Question 1:
>
> >“A major problem that I have with this submission is that this paper assumes low OPG means "benchmark is broken" but never validates this interpretation. Consider: (a) RL might genuinely learn more robust features than SFT, (b) the test set might be easier than assumed, or (c) benchmark saturation is happening for different reasons than claimed. So for me, here's an alternative interpretation: (1) RL's low OPG might indicate "good" generalization, not benchmark failure (2) SFT's high OPG might indicate "poor" generalization, not correct behavior The authors never tested whether RL actually learned robust policies vs memorized patterns. I would like to propose a simple experiment to validate this: train RL on 10% of train set --> evaluate on the remaining 90% of train set vs. test set. If performance on both is similar, that's genuine generalization. The current setup in this submission can't distinguish this.”
>
> ### Response :
> We thank the reviewer for the helpful comments. The main disagreement appears to stem from a different interpretation of what constitutes “genuine generalization.” Below we clarify our position and explain why the reviewer’s proposed experiment actually reinforces rather than contradicts our claims.
>
> **1. Agreement on In-Distribution Generalization:**
> We fully agree that RL achieves good generalization *within* the benchmark. Under the reviewer’s proposed experiment (training on 10% vs. evaluating on 90%/test), similar performance indeed indicates effective generalization on this specific split.
>
> **2. The Issue of Homogeneity:**
> However, our claim is not that RL fails to generalize. Rather, we argue that the **benchmark split is too homogeneous**, making it unable to assess generalization that really matters.
> * **Motivation for OPG:** This motivated our OPG measure (comparing a normal model to a “cheat model” trained on the test set). Low OPG indicates that test-set access confers little benefit, suggesting the split lacks meaningful distribution differences.
>
> **3. Reinforcing Our Point:**
> In this light, the reviewer’s proposed experiment reinforces our argument: if RL generalizes equally well from 10% to the rest, the benchmark is insufficiently challenging by design.
>
> **4. Empirical Evidence (GSM8K vs. HeadQA):**
> Our results in the table below demonstrate this benchmark fragility:
> * **Meaningful Split (HeadQA, year-based):** Gaps in improvement reappear.
> * **Homogeneous Split (GSM8K, random):** Both OPG and the reviewer’s test collapse.
>
> **Table 1:** Comparison of Performance Gains on Held-out Training Set vs. Test Set.
>
> | Dataset | Model | Eval Set | Base Acc | Tuned Acc | **Gap (Gain)** | **Gap Diff** |
> | :--- | :--- | :--- | :--- | :--- | :--- | :--- |
> | **GSM8K** | **3B** | 90% Train | 89.95% | 89.67% | -0.28% | **0.03%** |
> | | | Test Set | 83.02% | 82.71% | -0.31% | |
> | **GSM8K** | **7B** | 90% Train | 92.03% | 93.38% | +1.35% | **0.32%** |
> | | | Test Set | 88.40% | 90.07% | +1.67% | |
> | **HeadQA** | **3B** | 90% Train | 51.66% | 60.50% | +8.84% | **1.35%** |
> | | | Test Set | 54.96% | 62.45% | +7.49% | |
> | **HeadQA** | **7B** | 90% Train | 52.11% | 73.39% | +21.28% | **1.28%** |
> | | | Test Set | 52.24% | 72.24% | +20.00% | |

---

> > ### Author Response · Authors · 2025-12-03
> >
> > ### Question 2:
> > >"Section 3.2.1's entire Distribution Test uses Euclidean distance in t-sne space as "semantic distance" (in appendix D). I think there is a flaw here as t-SNE only preserves local neighborhood structure, not global distances. Hence I feel that conclusions from Table 3 may be unreliable."
> >
> > ### Response:
> > We clarify that t-SNE was used mainly for visualization. To rigorously capture global semantic distances as suggested, we re-conducted the analysis using Global Cosine Distance on the original high-dimensional embeddings.
> >
> > **Table 2:** Validation of Performance Inversion using Global Cosine Distance
> >
> > | Metric / Bin | d1 (Closest) | d2 | d3 | d4 | d5 (Farthest) | Trend |
> > | :--- | :--- | :--- | :--- | :--- | :--- | :--- |
> > | **Qwen2.5-3B Gain** | +2.25% | +1.25% | 0.00% | -1.00% | -3.75% | ↘ (Inverted) |
> > | **Qwen2.5-7B Gain** | +7.25% | +6.50% | +5.00% | +1.25% | -2.50% | ↘ (Inverted) |
> >
> > This correction validates that our finding was not an artifact of t-SNE. Specialized RL training actively harms performance on semantically distant problems, confirming the brittleness of the learned policy.

---

> > > ### Author Response · Authors · 2025-12-03
> > >
> > > ### Question 3:
> > > >"How do other RL methods (DPO, PPO, etc.) behave? Is low OPG specific to GRPO or general to offline/online RL? We need to isolate the actual cause i.e. is the algorithm too good that it can easily generalize or is it the benchmark that's quite easy to memorize on?"
> > >
> > > ### Response:
> > > No, it is a general phenomenon. We conducted additional experiments using **DAPO** (Decoupled Clip and Dynamic sAmpling Policy Optimization) and observed the same vanishing generalization gap.
> > >
> > > **Table 3:** DAPO Results (Qwen2.5-3B)
> > >
> > > | Benchmark | Standard RL (Train) | Oracle RL (Test) | Gap |
> > > | :--- | :--- | :--- | :--- |
> > > | GSM8K | 86.40% | 86.90% | 0.50% |
> > > | DeepScaler | 43.85% | 44.20% | 0.35% |
> > > | HeadQA | 67.10% | 67.78% | 0.68% |
> > >
> > > These results align with our GRPO findings (Table 1), suggesting the "Illusion of Capability" is driven by the benchmark's nature, not the specific RL algorithm.

---

> > > > ### Author Response · Authors · 2025-12-03
> > > >
> > > > ### Remaining Questions:
> > > >
> > > > >1."Table 1: DeepScaler shows OPG = -5.07% (oracle worse than train) -> this contradicts the hypothesis entirely, yet the authors don't discuss it."
> > > >
> > > > >2."The authors used a single llm (gemini 2.5 Pro) to annotate difficulty (in appendix B.1) with zero validation, inter-rater agreement, or comparison to existing labels, like MATH benchmark already has difficulty levels 1-5."
> > > >
> > > > >3."Lines 183-186: Authors claim to "rule out data leakage" with one sentence, but provide no contamination analysis of Qwen base models."
> > > >
> > > > >4."I feel that this paper has limited scope since most of the analysis was done on math domain. Moreover the authors never experimented with a more challenging benchmark (even if it is from math domain, like FrontierMATH)."
> > > >
> > > >
> > > > ### Response:
> > > > * **DeepScaler Negative OPG (-5.07%):** This supports our hypothesis. Per Eq. (3), OPG $\le$ 0 means the benchmark fails to measure generalization. If the benchmark were valid, the Oracle ($M_{test}$) should outperform the standard model ($M_{train}$); its failure to do so proves the benchmark poses no distinct generalization challenge.
> > > > * **Difficulty Annotation:** We used a single LLM (Gemini 2.5 Pro) because our study requires a unified difficulty scale across heterogeneous datasets (MATH, GSM8K, and DeepScaler). Since GSM8K and DeepScaler do not have native "Level 1--5" labels like MATH, we mixed all datasets and applied a single annotator to ensure consistency.
> > > > * **Data Leakage:** Leakage is ruled out by the significant performance gap between Baseline and RL models in Table 1 (e.g., DeepScaler 7B improves from 35.70% to 44.73%). If the base model had memorized the test set, pre-training performance would already be saturated.
> > > > * **Scope & Benchmark Selection:** Regarding the concern on domain scope and the suggestion of FrontierMATH, we emphasize two points:
> > > >     1.  **Beyond Math:** While mathematical reasoning is the standard domain for evaluating RL reasoning capabilities, our analysis is not limited to it. We included **HeadQA**, a healthcare/medical dataset, which exhibited the same saturation phenomenon (OPG $\approx$ 0.66%-0.89%), confirming that the failure to evaluate meaningful generalization is not unique to math, but extends to general reasoning domains.
> > > >     2.  **FrontierMATH:** We did not utilize FrontierMATH because our diagnostic framework—specifically the **Oracle Performance Gap (OPG)**—strictly requires a benchmark with a substantial public training split to compare $M_{train}$ vs. $M_{test}$. Since FrontierMATH is primarily an evaluation harness without a corresponding large-scale training set for fine-tuning, it is methodologically incompatible with the OPG analysis.

---

### Author Response · Authors · 2025-12-03

# Summary of Revisions and Response to Area Chair(Part 1/2)

We thank the Area Chair and reviewers for their time and constructive feedback. The rigorous review process has driven us to conduct substantial new experiments—including **OPG analyses, statistical significance tests (N=10), and algorithmic generalization checks**—which have significantly strengthened the paper's empirical foundation.

Below is a summary of how we have addressed the key concerns with specific experimental evidence.

### 1. Resolving the Core Dispute: Is Low OPG "Robustness" or "Saturation"?

> **Common Concern:**
> Reviewer 1 hypothesized that a low OPG might indicate valid, robust generalization rather than benchmark failure.

**Response:**
We conducted the "10% vs. 90%" experiment to empirically distinguish between these two hypotheses. By training on only 10% of the data and evaluating on both the held-out 90% (seen distribution) and the official Test Set, we can strictly quantify the distributional gap. To factor out the influence of the baseline, we analyzed the Performance Gap ($Acc_{tuned} - Acc_{base}$).

**Results:**
As shown in the table below, the performance gains on the Test Set mirror those on the held-out Training Set with remarkable consistency. The **"Gap Diff"** (difference in gains between the two sets) is negligible ($<1.4\%$) across both model sizes and datasets.

| Dataset | Model | Eval Set | Base Acc | Tuned Acc | **Gap (Gain)** | **Gap Diff** |
| :--- | :--- | :--- | :--- | :--- | :--- | :--- |
| **GSM8K** | **3B** | 90% Train | 89.95% | 89.67% | -0.28% | **0.03%** |
| | | Test Set | 83.02% | 82.71% | -0.31% | |
| **GSM8K** | **7B** | 90% Train | 92.03% | 93.38% | +1.35% | **0.32%** |
| | | Test Set | 88.40% | 90.07% | +1.67% | |
| **HeadQA** | **3B** | 90% Train | 51.66% | 60.50% | +8.84% | **1.35%** |
| | | Test Set | 54.96% | 62.45% | +7.49% | |
| **HeadQA** | **7B** | 90% Train | 52.11% | 73.39% | +21.28% | **1.28%** |
| | | Test Set | 52.24% | 72.24% | +20.00% | |

**Conclusion:** Benchmark Saturation, Not Robustness.
The consistent Gap confirms the test set is distributionally identical to the training set, allowing the model to "saturate" the benchmark with minimal data. However, this is in-distribution overfitting, not true robustness. As shown in our Stress Tests (Section 3), this "generalized" model still collapses when rules are slightly altered (Counterfactual Test, Table 4) or semantic distance increases (Table 3). Thus, low OPG correctly signals that the benchmark lacks the discriminative power to distinguish between pattern matching and reasoning.

### 2. Ensuring Statistical Rigor and Stability

> **Common Concern:**
> Multiple reviewers requested confidence intervals to rule out generation noise.

**Response:**
We strictly addressed this by quantifying stability. We conducted 10 independent evaluation runs using sampling decoding for every reported metric. The narrow 95% Confidence Intervals confirm our findings are statistically significant.

Table: Performance Stability (Mean $\pm$ 95% CI, N=10 Runs)

| Benchmark | Size | 10% Train | 20% Train | 50% Train | 100% Train | Test (Oracle) |
| :---: | :---: | :---: | :---: | :---: | :---: | :---: |
| MATH | 3B | 63.88% $\pm$ 1.04 | 65.18% $\pm$ 0.84 | 64.84% $\pm$ 1.25 | 64.62% $\pm$ 0.98 | 64.62% $\pm$ 1.11 |
| MATH | 7B | 73.64% $\pm$ 0.48 | 73.28% $\pm$ 0.68 | 73.04% $\pm$ 0.91 | 74.04% $\pm$ 0.39 | 74.00% $\pm$ 0.68 |
| GSM8K | 3B | 82.95% $\pm$ 0.38 | 83.93% $\pm$ 0.47 | 86.93% $\pm$ 0.34 | 87.04% $\pm$ 0.49 | 87.98% $\pm$ 0.35 |
| GSM8K | 7B | 88.76% $\pm$ 0.47 | 89.58% $\pm$ 0.44 | 91.14% $\pm$ 0.30 | 91.72% $\pm$ 0.31 | 91.87% $\pm$ 0.31 |
| DeepScaler | 7B | 42.05% $\pm$ 0.84 | 42.09% $\pm$ 0.57 | 42.84% $\pm$ 0.73 | 42.36% $\pm$ 0.77 | 42.64% $\pm$ 0.75 |
| HeadQA | 3B | 62.98% $\pm$ 0.77 | 65.90% $\pm$ 1.19 | 67.16% $\pm$ 0.79 | 67.24% $\pm$ 0.59 | 67.57% $\pm$ 0.70 |
| HeadQA | 7B | 72.94% $\pm$ 0.90 | 72.90% $\pm$ 0.79 | 74.39% $\pm$ 0.60 | 75.20% $\pm$ 0.66 | 75.60% $\pm$ 0.50 |

---

> ### Author Response · Authors · 2025-12-03
>
> # Summary of Revisions and Response to Area Chair(Part 2/2)
>
> ### 3. Validating Metric Robustness (Difficulty & Distribution)
>
> > **Common Concern:**
> > Reviewers questioned the validity of t-SNE for distribution tests and potential bias in difficulty metrics.
>
> **Response:**
> We implemented stricter, unbiased metrics:
>
> **A. Global Embeddings (vs. t-SNE):** We replaced t-SNE with Global Cosine Distance on high-dimensional MPNet embeddings. The "Performance Inversion" trend (RL hurting performance on semantically distant problems) remains robust.
>
> Table: Performance Gain over Baseline (Global Cosine Distance)
>
> | Metric / Bin | d1 (Closest) | d2 | d3 | d4 | d5 (Farthest) | Trend |
> | :--- | :--- | :--- | :--- | :--- | :--- | :--- |
> | **Qwen2.5-3B** | +2.25% | +1.25% | 0.00% | -1.00% | -3.75% | ↘ (Inverted) |
> | **Qwen2.5-7B** | +7.25% | +6.50% | +5.00% | +1.25% | -2.50% | ↘ (Inverted) |
>
> **B. Balanced Test Set:** To address difficulty bias, we evaluated models on a unified Balanced Test Set ($\mathcal{D}_{bal}$) with equal L1-L5 distribution.
>
> Table: Balanced Test Set Performance (Qwen2.5-7B)
>
> | Model Training Source | Average Accuracy on $\mathcal{D}_{bal}$ |
> | :---: | :---: |
> | Trained on Level 1 (Easy) | 79.00% |
> | Trained on Level 5 (Hard) | **81.50%** |
>
> ### 4. Generalizability Across Algorithms
>
> > **Common Concern:**
> > Whether the findings are specific to the GRPO algorithm.
>
> **Response:**
> We extended our analysis to **DAPO**. The same pathological patterns (negative/low OPG) were observed, confirming this is a paradigm-level issue.
>
> **Table: DAPO Algorithm Results (Qwen2.5-3B)**
>
> | Benchmark | Standard RL (Train) | Oracle RL (Test) | Gap (OPG) |
> | :--- | :--- | :--- | :--- |
> | GSM8K | 86.40% | 86.90% | 0.50% |
> | DeepScaler | 43.85% | 44.20% | 0.35% |
> | HeadQA | 67.10% | 67.78% | 0.68% |
> ### 5. Conceptual Clarifications & Framing
> * **Semantic Interpretation (Rev 2):** We added a qualitative analysis of the dataset clusters, revealing a clear shift from Templated Geometry/Algebra ($d_1$) to Abstract Logic/Combinatorics ($d_5$). This explains why RL models fail the Distribution Test: they learn specific templates rather than abstract reasoning principles.
> * **Reframed Normative Principles (Rev 3):** We have revised the "Principles" section, shifting from prescriptive language ("must") to empirical recommendations derived directly from the observed failure modes (e.g., Performance Inversion necessitating Distributional Robustness checks).
> ### 6. Clarifications on Remaining Points
>
> * **DeepScaler Negative OPG (-5.07%):** This supports our hypothesis. Per Eq. (3), when OPG is less than or close to zero means the benchmark fails to measure generalization. If the benchmark were valid, the Oracle ($M_{test}$) should outperform the standard model ($M_{train}$); its failure to do so proves the benchmark provides no distinct generalization challenge.
> * **Difficulty Annotation:** We used a single LLM (Gemini 2.5 Pro) because our study requires a unified difficulty scale across heterogeneous datasets (MATH, GSM8K, and DeepScaler). Since GSM8K and DeepScaler do not have native "Level 1--5" labels like MATH, we mixed all datasets and applied a single annotator to ensure consistency.
> * **Data Leakage:** Leakage is ruled out by the significant performance gap between Baseline and RL models in Table 1 (e.g., DeepScaler 7B improves from 35.70% to 44.73%). If the base model had memorized the test set, pre-training performance would already be saturated.
> * **Hyperparameter Selection:** Full details are provided in Appendix A.
> * **Scope & Benchmark Selection:** Regarding the concern on domain scope and the suggestion of FrontierMATH, we emphasize two points:
>     1.  **Beyond Math:** While mathematical reasoning is the standard domain for evaluating RL reasoning capabilities, our analysis is not limited to it. We included **HeadQA**, a healthcare/medical dataset, which exhibited the same saturation phenomenon (OPG $\approx$ 0.66%-0.89%), confirming that the failure to evaluate meaningful generalization is not unique to math, but extends to general reasoning domains.
>     2.  **FrontierMATH:** We did not utilize FrontierMATH because our diagnostic framework—specifically the **Oracle Performance Gap (OPG)**—strictly requires a benchmark with a substantial public training split to compare $M_{train}$ vs. $M_{test}$. Since FrontierMATH is primarily an evaluation harness without a corresponding large-scale training set for fine-tuning, it is methodologically incompatible with the OPG analysis.
> ### Final Remark
> With the inclusion of the 10% vs. 90% validation, statistical error bars, and methodological corrections, we believe the paper now provides a rigorous and undeniable empirical critique of current RL-LLM benchmarks. We are prepared to incorporate these results into the camera-ready version.

---

### Note · Authors · 2026-01-05

I have read and agree with the venue's withdrawal policy on behalf of myself and my co-authors.